# T Cell Receptor Sequences Amplified during Severe COVID-19 and Multisystem Inflammatory Syndrome in Children Mimic SARS-CoV-2, Its Bacterial Co-Infections and Host Autoantigens

**DOI:** 10.3390/ijms24021335

**Published:** 2023-01-10

**Authors:** Robert Root-Bernstein, Elizabeth Churchill, Shelby Oliverio

**Affiliations:** 1Department of Physiology, Michigan State University, East Lansing, MI 48824, USA; 2School of Health Sciences, George Washington University, Washington, DC 20052, USA

**Keywords:** COVID-19, Kawasaki disease, autoimmune disease, T cell receptor sequences, molecular mimicry, antigenic complementarity, anti-idiotype, idiotypic network, bystander activation, similarity, autoantigens

## Abstract

Published hypervariable region V-beta T cell receptor (TCR) sequences were collected from people with severe COVID-19 characterized by having various autoimmune complications, including blood coagulopathies and cardiac autoimmunity, as well as from patients diagnosed with the Kawasaki disease (KD)-like multisystem inflammatory syndrome in children (MIS-C). These were compared with comparable published v-beta TCR sequences from people diagnosed with KD and from healthy individuals. Since TCR V-beta sequences are supposed to be *complementary to* antigens that induce clonal expansion, it was surprising that only a quarter of the TCR sequences derived from severe COVID-19 and MIS-C patients *mimicked* SARS-CoV-2 proteins. Thirty percent of the KD-derived TCR mimicked coronaviruses other than SARS-CoV-2. In contrast, only three percent of the TCR sequences from healthy individuals and those diagnosed with autoimmune myocarditis displayed similarities to any coronavirus. In each disease, significant increases were found in the amount of TCRs from healthy individuals mimicking specific bacterial co-infections (especially *Enterococcus faecium*, *Staphylococcal* and *Streptococcal* antigens) and host autoantigens targeted by autoimmune diseases (especially myosin, collagen, phospholipid-associated proteins, and blood coagulation proteins). Theoretical explanations for these surprising observations and implications to unravel the causes of autoimmune diseases are explored.

## 1. Introduction

The expansion of specific T cell receptor (TCR) clones is non-random during the disease process, driven by the binding of antigens to the receptors, and has been well-characterized in many diseases, including autoimmune diseases (e.g., [1,2,3,4,5,6,7]). However, the relationship between TCR sequence expansion in particular autoimmune diseases to the peptide sequences expressed by the antigens to which they have been amplified has rarely been explored [1,2,8,9]. One reason for the absence of such analyses is the well-founded assumption that the V-beta regions of the TCR are *complementary to* the antigens that induce expansion of the relevant T cell clones. Since there is, at present, no well-founded algorithm or theory to predict the antigen sequence from the TCR sequence (or vice versa), there is no *a priori* reason within standard immunological theory to identify that a given sequences of a TCR and an antigen will display any predictable sequence relationship. Since these sequences are presumed to be complementary, there is certainly no reason within current immunological theory to think that TCR and antigen sequences are very similar or identical.

It therefore comes as a surprise that a handful of recent studies have demonstrated that, in at least some autoimmune diseases, a triangle of *mimicry* relationships—not *complementary* relationships—exists between the V-beta TCR sequences amplified by the host in response to infection, to the infectious triggers of the disease, and to the autoantigen targets of autoimmunity. For example, in type 1 diabetes, TCR sequences mimic putative triggers of the disease including coxsackieviruses, cytomegalovirus (CMV), *Clostridia* and *M. tuberculosis*, and they do so at a statistically significantly higher rate than TCRs from healthy individuals [8]. These amplified TCR sequences also mimic self-antigens that are targets of T cells in T1DM, such as insulin, glutamic acid decarboxylase and the insulin receptor and these TCRs are recognized as autoantigens themselves by T1DM autoantibodies [9]. Similarly, the TCR sequences amplified in Crohn’s disease mimic its putative triggers, specifically *Enterobacteriaceae* such as *E. coli*, *Corynebacteria*, *Salmonella*, *Candida*, *Pseudomonas* species and atypical *Mycobacteria* [8], microbes that in turn mimic the host autoantigens targeted by the disease [10,11,12]. Additionally, in acquired immunodeficiency syndrome (AIDS), people with full-blown AIDS are often characterized by the presence of lymphocytotoxic autoantibodies (LCTA) [13,14,15,16,17] targeting TCRs that mimic human immunodeficiency virus type 1 (HIV-1) antigens [18,19,20,21]. 

This paper explores whether the pattern of TCR–host–microbe associations established in diabetes, Crohn’s disease and HIV-related autoimmunity also characterizes some of the autoimmune complications associated with the recent COVID-19 pandemic, such as coagulopathies, myocardial autoimmunity and multisystem inflammatory syndrome in children (MIS-C). COVID-19 is a disease caused by the SARS-CoV-2 virus. Most cases resolve without long-term complications, but autoimmune diseases often follow serious and severe cases and are a probable cause of what has been called “long COVID” [22,23,24,25,26,27]. Long COVID is much more frequent (25%) among people who have been admitted to intensive care (43.1%) than those hospitalized (23.5%) or those never hospitalized (5.7%) [28] and symptoms can involve systems ranging from thyroid disfunction to neurological complications. Additionally, among the most common long-term complications observed in long COVID patients are autoimmune coagulopathies, such as thrombocytopenia and microclotting targeting a range of host antigens, including cardiolipin (CL), platelet factor 4 (PF4), beta 2 glycoprotein I (β2GPI), various clotting factors, collagens, phosphatases and phospholipids [29,30,31,32]. While people vaccinated against COVID-19 and mild cases of COVID-19 have no increased risk of autoimmune coagulopathies, 10–15% of hospitalized patients, 25% of critically ill COVID-19 patients and up to 48% of intensive care patients [33,34,35,36,37,38,39,40] develop autoimmune coagulopathies. 

Various forms of autoimmune heart disease also characterize long COVID, targeting host antigens (some shared with coagulopathies), such as myosin, actin, laminin, collagens and CL [41,42,43]. An average of eighteen percent (range 8 to 64%) of COVID-19 patients across the entire range of disease severity, including (rarely) previously healthy young athletes, experience cardiac injury as measured by magnetic resonance imaging and increased troponin during recovery from their illness (reviewed in [44]). Autoimmune myocarditis is also the most common post-acute COVID-19 complication among children and adolescents [45]. MIS-C, an autoimmune [46,47,48,49,50,51,52,53,54,55] Kawasaki disease-like syndrome that follows SARS-CoV-2 infection by several weeks [51,52], also occurs rarely among post-infectious complications seen in children with severe cases of SARS-CoV-2. MIS-C is characterized by vasculitis, cardiomyopathy and various other symptoms associated with hyperinflammation, such as sepsis and cytokine storm. T cell receptor sequencing has been performed on all these groups, including Kawasaki disease (KD) (see sources referenced in the Section 4) providing the possibility of exploring whether these TCR sequences unexpectedly mimic SARS-CoV-2. Since the cause or causes of KD are unknown and range from viruses to bacteria to vaccines [53,54,55,56,57], and since KD was discovered long before SARS-CoV-2 was identified, KD TCRs provide good control for MIS-C TCRs.

We also investigated whether human viruses and bacteria other than SARS-CoV-2 mimic the TCR sequences expanded during severe COVID-19. The rationale for this broader similarity search was two-fold. One was the necessity of having a range of appropriate controls. The other was that people infected with SARS-CoV-2 who experience no or mild symptoms very rarely develop additional viral, bacterial or fungal infections and very rarely develop autoimmune complications [28,33,34,35,36,37,38,39,40], whereas those who develop severe or fatal COVID-19 almost always develop additional viral, bacterial or fungal infections. The most common secondary viral infections include adenoviruses and influenza viruses while the most common bacterial infections include *Mycoplasma pneumoniae*, *Staphylococcus aureus*, *Legionella pneumophila*, *Streptococcus pneumoniae*, *Haemophilus* and *Klebsiella* species as well as *Mycobacterium tuberculosis* coinfections [58]. The bacterial infections in particular are found in up to half of hospitalized patients and the majority of those admitted to intensive care [59,60,61]. Some of these bacteria are also associated with an increased risk of autoimmune myo- and endocarditis in COVID-19 including *Streptococcus mitis* and *oralis*, *Enterococcus faecalis*, *Staphylococcus aureus*, or coagulase-negative *staphylococci* [62,63,64]. *Enterococcus* infections are particularly associated with the risk of hospitalization, admission to intensive care, and the increased risk of mortality in COVID-19 patients [65,66,67]. These bacterial infectious would therefore be expected to have been present in a significant proportion of severe COVID-19 patients from which the TCRs utilized in this study were derived and because of the severity of their disease, these patients would also be at the highest risk for developing autoimmune complications [28,33,34,35,36,37,38,39,40,45]. Thus, some of the TCR clones expanded during their autoimmune disease might reflect a response to these additional infections and this possibility must be taken into account in evaluating any increased rate of antigen mimicry by TCRs during the disease process.

In addition to the mimicry of microbial antigens, previous studies [8,9,10,11] have demonstrated that every human TCR sequence mimics some set of human antigens as well so that a baseline probability of such mimicry must be established in order to recognize significant differences associated with COVID-19 autoimmune diseases. Thus, part of this study involved establishing baseline probabilities that TCR sequences from healthy individuals mimic the range of bacterial, viral and human antigens examined. The resulting statistical studies are reported here. An investigation of the specific similarities to infectious agents using sets of TCR sequences from individual patients was also carried out if the sources of the TCR sequences made the appropriate information available. 

Briefly, we found that some TCR sequences from COVID-19 patients with severe disease and/or autoimmune sequelae do mimic SARS-CoV-2 at an unexpectedly high rate and also mimic several common bacteria and viruses known to complicate this viral infection such as *Streptococci*, *Staphylococci* and *Enterococcus faecium*. These TCR sequences also mimic at significantly increased rates some of the molecular host autoantigens that are known to be targets of these COVID-19-associated autoimmune diseases, such as myosin, collagen, phosphatases, phospholipases, and olfactory receptors. The Section 3 addresses the possible mechanisms by which this surprising triangular relationship of similarities shared by host autoantigens, TCR sequences, and microbial antigens may have evolved and possible functions of this mimicry triangle in the induction of autoimmune diseases.

## 2. Results

### 2.1. Statistical Analysis of COVID-19 TCR Sequence Similarity to Microbial Sequences

Initial studies were performed to determine the frequency with which 325 TCR sequences from healthy individuals mimicked a range of approximately 40 viruses and 40 bacteria that commonly infect human beings. The sources of these TCR sequences are provided in the Section 4. Significant similarity was defined as a TCR sequence sharing at least six amino acid identities (with a pair of similar amino acids counting as a single identity) over a sequence of ten amino acids or five consecutive identities, criteria that has been tested experimentally and shown to predict antigenic cross-reactivity with about 85% accuracy [68,69,70,71,72,73]. One notable result is that every TCR sequence significantly mimics some small set of viral and/or bacterial antigens, which is consistent with previous studies [8,9,10,11,18,19,20,21,22]. All of the TCR also mimicked multiple human proteins with a very high degree of similarity and 39 proteins known to be targets of autoimmune coagulopathies, cardiopathies or vasculopathies were chosen for analysis. For the purposes of the present study, it was assumed that these virus, bacteria and human antigen similarities to TCR sequences arise by chance providing a baseline of the probability that any given TCR sequence may randomly mimic any given protein from these sources. 

The results of the study of the TCR from healthy individuals were compared to the TCR sequence similarities derived from hospitalized individuals with moderate-to-severe COVID-19 (198 TCR), MIS-C patients (150 TCR), and patients diagnosed prior to COVID-19 with Kawasaki disease (KD) (69 TCR) (sources again are provided in the Section 4). Significant differences between the frequency of similarities found among the healthy and disease TCRs was determined initially using a chi-squared analysis supplemented by Bonferroni corrections because each TCR sequence was compared with multiple viruses and bacteria. A significant correction at the *p* < 0.05 level after Bonferroni corrections required that the chi-squared *p* value be less than 0.002. Values near or below this value are bolded in figures that follow for ease of identification. 

Figure 1 compares healthy TCRs with moderate-to-severe COVID-19 TCRs and MIS-C TCRs in terms of their virus protein mimicry. Notably, 24% of COVID-19 TCRs and 27% of MIS-C TCRs mimic coronaviruses. Among hospitalized COVID-19 patients, most of the mimicry involved SARS-CoV-2 proteins while, interestingly, the majority of similarities for MIS-C patients were to other human coronaviruses. Bat coronaviruses appeared very frequently but were not included in the mimicry counts. The mimicry with human coronaviruses is the only significant deviation from the “normal” distribution of similarities found for the healthy TCR set for the COVID-19 set. The MIS-C TCRs also demonstrated significant, or near-significant, increases in similarities to antigens of herpes viruses 1 and 2 and parainfluenza virus with a near-significant decrease in similarities to reoviruses. In short, people with serious COVID-19 infections display very significant increases in TCRs that mimic SARS-CoV-2 and in MIS-C patients, as well as herpes viruses and the parainfluenza virus. Adenovirus mimicry was also increased, but not significantly, in both disease groups compared with the TCRs from healthy individuals.

Figure 2 compares 325 healthy TCRs with 198 moderate-to-severe COVID-19 TCRs and 150 MIS-C TCRs in terms of their bacteria protein mimicry. These results are not quite as “clean” as the virus data, which is not surprising given that severe COVID-19 patients are likely to be infected with a range of possible bacteria but, by definition, are all infected with one common virus. Nonetheless, it is notable that atypical mycobacterial proteins and *Enterococcus faecium* proteins display significantly increased mimicry with both COVID-19 and MIS-C TCRs suggesting that both of these bacteria may be important factors in COVID-19 severity for a significant number of patients. *E. coli*, *Salmonella*, *Staphylococcus* and *Streptococcus* mimicry was also increased among COVID-19 TCRs, reflecting the observation that these infections are also commonly observed among moderate-to-severe cases (see Introduction). It is notable that these latter bacteria do not appear among the TCR significant mimics in MIS-C patients, perhaps suggesting that MIS-C is a result of specific interactions between SARS-CoV-2 and *Mycobacteria* and/or *E. faecium* while the range of autoimmune complications seen in the broader COVID-19 population is a reflection of the broader set of bacterial co-infections these patients experience. It should again be emphasized that the observation that statistically significant increases in TCR mimicry of bacteria in COVID-19 and MIS-C is associated only with select bacteria known to have high rates of infection among these groups.

Figure 3 compares 325 healthy TCRs with 198 moderate-to-severe COVID-19 TCRs and 150 MIS-C TCRs in terms of their human protein mimicry. Every human TCR mimics some range of human proteins [8,9] so that, in a sense, the immune system represents a “body double” of the proteome that can intercept threats to the host. In the case of COVID-19, TCRs mimicking human leukocyte antigens (HLA), Toll-like receptors (TLR), olfactory receptors and phospholipases are significantly increased compared with TCRs from healthy individuals. These targets may indicate that autoimmunity involves the dysregulation of immunity (HLA and TLR) with olfactory receptors (anosmia) and anti-phospholipid syndrome (APS) as the most common results. Increases were also observed in the mimicry of other proteins that did not reach statistical significance in this study, such as cardiomyopathy-associated proteins and thrombospondin, which may indicate that subsets of the COVID-19 group experienced autoimmunity related to these targets. Unfortunately, the autoimmune complications were not listed for any of the COVID-19 individuals so that it was not possible to provide a breakdown or sub-analysis. In contrast, the TCRs of MIS-C patients showed significantly increased similarities to collagen and myosin, as might be expected in autoimmune cardiopathies; heparin-related proteins such as heparin sulfate sulfotransferases and phosphatases, which may relate to MIS-C coagulopathies; and glutamate receptors, which may impact vascular and muscle function. MIS-C TCRs shared only one enhanced set of similarities with COVID-19 TCRs, which was to mimic TLR, again suggesting autoimmunity involves dysregulation within the immune system itself. These results suggest that the average severe COVID-19 patient experiences a different set of autoimmune targets than the typical MIS-C patient. 

### 2.2. Analyzing TCR Sets from Individual Patients

Where sequenced sets of TCRs for individual patients were available, a more in-depth analysis of the relationship between the triangle of viral, bacterial and host protein mimicry was possible. A very limited example consisting of only four TCRs from a single surviving COVID-19 patient from a study by Schultheiss et al. [74] is presented in Figure 4 and three additional more extensive sets are provided in Appendix A. What is notable about each of these sets is that, as expected from the statistical results reported in Figure 1, some of the TCRs significantly mimic SARS-CoV-2 sharing six or more identical amino acids in a series of ten or five identical amino acids in a row (and often some additional conserved amino acid substitutions). Such sequences have a high rate of probability of demonstrating cross-reactivity in antibody studies [68,69,70,71,72,73]. Some of the TCRs also significantly mimic bacterial infections associated with severe COVID-19, such as *Streptococci*, *Staphylococci*, *E. coli*, *Pseudomonas aeruginosa*, *Haemophilus influenzae*, and *Acinetobacter baumannii*. Although not all of these bacterial similarities rose to significance in the statistical study (Figure 2), it may be possible that they represent co-infections in the particular individual. Additionally, many of the TCRs also mimic human proteins targeted by autoimmune processes during severe COVID-19, such as olfactory and taste receptors, phosphatases targeted in APS, blood proteins associated with coagulopathies, and heart-related proteins such as laminins, collagens and myosin. Again, although not all of these similarities rose to statistical significance for the COVID-19 population, they may indicate unique targets for specific individuals. Figure 3 also displays multiple similarities between one of the expanded COVID-19-related TCRs and mucins, which function as essential antibacterial proteins, perhaps indicating that various aspects of immune function are targets of autoimmunity in some patients. The fact that so many of these proteins show up in sets of TCRs from individual patients but not in the statistical results summarized in Figure 3 is likely due to the fact that each of the patients illustrated in Figure 4 and Appendix A has a unique distribution of human protein matches, diluting their statistical significance across the population of COVID-19 TCR sequences. This dilution effect should not blind us to the possibility that the individualized analysis of TCR mimicry may provide more nuanced insights into individual autoimmune complications. 

### 2.3. Comparing TCR Mimicry Distributions in MIS-C and KD Patients

A set of analyses similar to those carried out for COVID-19 and MIS-C was also carried out for the TCRs from patients diagnosed with KD prior to the COVID-19 pandemic. These analyses permit us to address the ongoing question of how similar MIS-C and KD are [75,76,77,78] from a new perspective and perhaps shed light on the perplexing problem of the etiology of KD. Figure 5 summarizes those comparisons with regard to TCR–virus similarities. While TCRs mimicking coronaviruses are statistically significantly increased in both KD and MIS-C patients, none of the KD patient TCRs mimicked SARS-CoV-2, instead displaying similarities to more common coronaviruses. This phenomenon is more clearly illustrated in the examples in Appendix B where two individual KD patient TCR sets are displayed in detail. KD TCRs also differed from MIS-C TCRs in significantly mimicking reoviruses rather than the rotaviruses and herpes viruses that were not found for MIS-C TCRs. These results may either indicate that these viruses can synergize with coronaviruses to trigger these autoimmune consequences or are alternative triggers in and of themselves that are common enough to rise to statistical significance.

Figure 6 suggests similarly that KD and MIS-C may differ in the types of bacteria that are involved in disease pathogenesis. TCRs mimicking pathogenic *Clostridia*, *E. coli*, *Mycobacteria*, *Salmonella* and *Staphylococci* all rose to statistical significance in KD. MIS-C was also characterized by TCRs mimicking *Mycobacteria* but none of the other bacteria. Instead, MIS-C TCRs mimicked *Enterococcus faecium*. 

Figure 7A,B and Figure 8 and Appendix B provide further information concerning the individual distributions of significant TCR mimicry to individual viruses and bacteria, emphasizing the point that while coronaviruses are the viruses most often mimicked in these patients, they are not universally mimicked in KD patients for whom rotaviruses and herpes viruses are also very common; and similarly, while *Enterococcus faecium* is the most common bacterium found in MIS-C TCR mimicry, not every MIS-C patient displays this mimicry, some displaying mimicry to other bacteria such as *Staphylococci*, *Streptococci*, and (in the case of KD) *Clostridia* instead. This diversity suggests that while only a very limited range of bacteria appear consistently within the TCR mimicry displayed by expanded lymphocytes in KD and MIS-C, it may not be possible to identify a single virus or bacterium that is both necessary and sufficient to trigger these autoimmune syndromes. On the other hand, the sets of TCR similarities to viruses and bacteria displayed by every KD and MIS-C patient strongly suggest that expanded TCRs always mimic at least one virus and one bacterium that is among those with significantly increased frequency in Figure 1, Figure 2, Figure 4 and Figure 5.

Finally, it is important to note that the human proteins mimicked by KD TCR did not differ significantly from those mimicked by MIS-C, which helps to explain their many shared symptoms. Because no significant differences were found, a figure illustrating this fact was not deemed of sufficient interest to include here and the data are, therefore, not displayed.

## 3. Discussion

### 3.1. Summary of Results

To summarize, as hypothesized in the Introduction, TCR sequences from hospitalized COVID-19 patients, MIS-C patients and KD patients each displayed significantly increased rates of mimicry to viruses and bacteria associated with their diseases compared with the distributions of such mimics calculated from the TCRs of healthy individuals. COVID-19 TCRs and MIS-C TCRs display unusually high rates of mimicry for SARS-CoV-2 proteins (around 25%), while KD TCRs displayed correspondingly high rates of mimicry for non-SARS coronaviruses compared with a mimicry rate for coronaviruses of only 3% among randomly chosen TCRs from healthy individuals. Rotavirus mimicry was also significantly increased in MIS-C TCRs, while increased herpes virus and parainfluenza mimicry accompanied KD TCRs. A significant association between COVID-19 infection and primary HSV infection or reactivation has been observed [81,82] and the combination of SARS-CoV-2 and herpes simplex can be fatal in children [83]. However, herpes simplex infections are very rare among MIS-C patients [84] and there appear to be no reports of parainfluenza complicating SARS-CoV-2 in MIS-C patients. Thus, the reasons for the significantly increased percentage of TCRs mimicking herpes simplex and parainfluenza antigens is not immediately evident. 

As for KD, coronaviruses, parainfluenza viruses and adenoviruses, each of which are implicated in our results, have also been identified as possible triggers for the disease [85,86,87,88,89,90,91,92,93]. However, antibody studies have not yet validated these findings for larger groups of KD patients. While one study found evidence of increased IgG and IgM antibodies to adenovirus type 2 in the majority of KD patients, no increases in herpes types 1 or 2, varicella zoster virus or CMV were found [94]. A similar study found no significant differences in the seropositive rates of antibodies to EBV, cytomegalovirus, herpes simplex virus and herpes zoster virus comparing KD patients with healthy controls [95]. EBV was also ruled out as a possible cause of KD in Hawaiian patients [96]. However, attempts to link these infections to the incidence of KD by means of epidemiological studies have failed to find any temporal correlation with very inconsistent results characterizing these studies in terms of correlations between other viruses, such as influenza, RSV, bocaviruses, enteroviruses and the temporal onset of KD [97,98,99,100]. Notably, rotaviruses, which are implicated in our TCR study, do not appear to have been studied with regard to KD. The failure to identify any particular causal agent with regularity other than coronaviruses may be due to the possibility that KD results from combined infections. In some cases, the viral infection has been complicated by concurrent bacterial infections. Johnson and Azimi [86] documented a case of KD diagnosed with parainfluenza type 3 virus infection and *Klebsiella pneumoniae*. 

Overall, it seems logical to focus on the fact that coronaviruses are common in severe COVID-19, MIS-C and KD but the presence of other viruses in MIS-C and KD may be important clues to possible etiologies involving combined infections.

Statistically significant, or near-significant, increases in the TCR mimicry of bacteria associated with severe and fatal COVID-19 were also found in our study, particularly for *Mycobacteria* (particularly atypical species), *Enterococcus faecium*, *Salmonella*, *Staphylococci* and *Streptococci.* These are all among the most-commonly diagnosed infections complicating SARS-CoV-2 infections (see Introduction) which suggests that TCR mimicry of their antigens is not due to chance. Significant increases in mimicry of MIS-C TCRs for *Enterococcus faecium* and *Mycobacteria* were also observed suggesting that these bacteria may play an especially important role in promoting cardiac and vascular complications in SARS-CoV-2-infected patients. While KD TCRs also displayed significantly increased mimicry with *Mycobacteria*, they notably also displayed significant increases for pathogenic *Clostridia*, *Salmonella* and *Staphylococci*. Thus, KD etiology may involve not only non-SARS coronaviruses but a different set of bacterial cofactor infections that result in a similar, but not identical, syndrome to MIS-C. Taken together, the sets of virus and bacteria mimicry of TCRs in severe COVID-19, MIS-C and KD suggest that autoimmune complications are multifactorial [69,70,101]. This conjecture seems to be supported by the analysis of TCR sets from individuals provided in the Section 2 and Appendix A and Appendix B. 

Both statistical studies and analyses of the sets of individual TCRs demonstrate that TCR sequences from each disease group also mimic human proteins associated as possible autoantigenic targets of their disease, and they do so at significantly increased rates compared with the distribution of such mimics calculated from the TCRs of healthy individuals. For COVID-19, these include human leukocyte antigens (HLA), both type 1 and 2; Toll-like receptors (TLR); phospholipases; and olfactory receptors with non-significant trends towards increased actin-related proteins, glutamate receptors, blood factors, platelet-related proteins including thrombospondin, and renin (angiotensinogenase). If it were possible to identify specific groups of COVID-19 patients by their particular autoimmune disease (coagulopathies versus cardiopathies versus anosmia, etc.) perhaps these non-significant trends would associate more strongly with particular types of autoimmunity. The greater uniformity of autoimmune symptoms in MIS-C and KD was reflected in a greater synchrony of TCR mimics of human proteins, collagens, myosins and glutamate receptors all being possible targets of smooth and cardiac muscle autoimmunity [66,69,70,71,102] and phosphatases being possible targets in anti-phospholipid syndrome (APS). Non-significant trends towards increased TCR mimicry to adrenergic receptors, complement proteins and endothelin-converting enzyme were also apparent, which could also contribute to MIS-C and KD autoimmune pathologies. 

Perhaps the most important result of this study is illustrated in the case studies of the virus, bacterium and human protein mimicry of sets of TCRs from individual patients. These clearly demonstrate that the viruses and bacteria display significant similarities not only to the TCRs but also to specific human proteins associated with their autoimmune pathologies. Thus, as has been previously demonstrated [71], *Streptococcal* proteins mimic myosins as do other bacteria such as *Staphylococci* and *Enterococcus faecium* [62,63,64] and this fact is evident in many of the individual sets of TCRs analyzed here and in the Appendix A and Appendix B. These bacteria can also induce antibodies that recognize a range of blood proteins, including cardiolipin, b2GPI, platelet factor 4, and other coagulation factors, as antigens [69,70]. Similarly, coronaviruses such as SARS-CoV-2 have been demonstrated to induce antibodies that cross-react with a range of human proteins including von Willebrand factor, phosphodiesterases, phospholipids [69,70] and possibly platelet factor 4 [69,70,103], as well as myosin, actin, collagen and the beta 2 adrenergic receptor [104,105,106] (Figure 9). Thus, the range of autoantigens that are targets of autoimmune diseases that complicate COVID-19 almost certainly require combinations of bacteria with one or more viruses [69,70]. These combinations of coronaviruses with different bacteria (and possible other viruses as well) might explain why individuals develop specific autoimmune complications as a result of COVID-19, MIS-C or KD and why the specific targets of that autoimmunity may vary from individual to individual depending on the specific sets of human proteins and TCRs that the viral and bacterial antigens mimic. 

### 3.2. Explaining the TCR Mimicry of Pathogen and Host Antigens

The expansion of TCRs that mimic specific combinations of viruses and bacteria, in severe COVID-19, MIS-C and KD raises a series of interrelated questions concerning the mechanism(s) behind this mimicry and its function within the context of autoimmunity. In particular, it seems very odd that TCR sequences expanded in response to a SARS-CoV-2 infection should mimic viral antigens. Equally odd is the observation that many of these expanded TCR sequences specifically mimic infectious agents known to complicate COVID-19, such as *Streptococci*, *Staphylococci*, and *Enterococci*. Why these bacteria and not others? The same puzzles attend the mimicry of expanded TCRs in KD for coronaviruses and herpes viruses and *Enterococci*. The fact that these expanded TCR sequences also mimic host proteins such as myosin, collagen, olfactory receptors and blood proteins that are targets of autoimmunity in these diseases also poses a series of conundrums. 

There are several theories of autoimmune disease initiation by which the results reported here might be explained, which include the molecular mimicry theory, anti-idiotype theory, bystander activation theory and complementary antigen theory, each of which is supported by extensive data related to autoimmune myocarditis [107,108] and therefore are particularly relevant in the present context. 

The dominant theory of autoimmune disease for many decades has been the molecular mimicry theory which posits that autoimmune diseases result when antigens from an infection agent trigger an immune response from the host that cross-reacts with autoantigens that mimic the pathogen’s antigen [109,110,111,112] (Figure 10). In essence, a virus, such as SARS-CoV-2, mimics a self-protein on a host cell. The immune system responds by activating T or B cells that express T cell receptors (TCR) and/or antibodies (shown here for simplicity) that are complementary to the viral antigens. Because of the mimicry between the viral antigens and the self-protein, some of the resulting TCRs and/or antibodies may target host cells expressing these self-proteins, resulting in autoimmune disease. Thus, molecular mimicry theory does not predict the expansion of TCRs (or antibodies) that mimic SARS-CoV-2. Thus, while mimicry is clearly present in the results reported here, the mimicry found here is of a completely different nature than that predicted by the molecular mimicry theory. Rather than the pathogen-derived antigen mimicking the host autoantigen and the immune response being complementary to both, here we report that the immune response also mimics the pathogen-derived antigen and host autoantigens. This sort of mimicry is of a novel sort. Additionally, molecular mimicry theory does not provide any explanation for why mimicry to possible bacterial co-infections should appear among the same sets of TCRs or antibodies.

A second possible explanation for the results reported here is the anti-idiotype theory of autoimmune disease. According to this theory [113,114,115], viruses utilize specific host receptors (angiotensin-converting enzyme 2, in the case of SARS-CoV-2 [116]) inducing an immune response that mimics the receptor. If this idiotypic immune response goes on to provoke an anti-idiotype response, then the resulting TCRs (or antibodies) would attack the same host target as the virus (Figure 11). This theory might be applied to our results as follows. Since the vast majority of the COVID-19 TCR sequences utilized in this study were derived from patients who survived their disease, the distribution of these TCRs represents the post-acute phase of their immune response and may therefore represent a mixture of idiotypic and anti-idiotypic responses to SARS-CoV-2 infection. One would therefore expect that some of the expanded TCRs would be anti-idiotypic ones that would mimic SARS-CoV-2 antigens and target the ACE-2 receptor. So far, so good. However, a number of limitations make the anti-idiotype theory an unlikely one for explaining COVID-19 autoimmune disease. One limitation is that ACE2 does not appear to be a primary target of autoimmunity in COVID-19, and certainly not in COVID-19 myocarditis, coagulopathies or anosmia/dysgeusia. Additionally, the anti-idiotype theory predicts that the antigens of the virus triggering the disease should be complementary to host antigens attacked in the autoimmune disease rather than mimicking them, as is the case reported here. Additionally, as with the molecular mimicry theory, the anti-idiotype theory cannot explain the similarities that are observed by expanded TCRs to bacterial infections associated with severe COVID-19. Thus, the observation that the TCRs expanded in COVID-19 mimic with significant probability the antigens of bacterial co-infections highly associated as co-infections or super-infections among severe COVID-19 patients remains unexplained by this theory. Finally, one limitation that is general to both the anti-idiotype theory and the molecular mimicry theory is that neither explain why only some people go on to develop autoimmune disease while other people infected with the same microbe do not produce sufficient mimics or anti-idiotypes to produce autoimmune disease.

A third possible explanation for the results reported here provides a possible explanation for why anti-idiotypes develop among some autoimmune disease patients and not among most people infected with SARS-CoV-2. Autoimmune disease may require both molecular mimicry of the pathogen for host autoantigens as well as a bystander infection (or infections) to produce a hyperinflammatory environment in which “self” tolerance can be abrogated and anti-idiotype immune responses initiated [116,117,118] (Figure 12). Idiotype–anti-idiotype antibodies or TCRs would result from the mechanism described by the anti-idiotype theory but be enabled by the secondary infection. Notably, the bystander activation theory does not require that there be any specific relationship between the bystander infection and host autoantigens or between the primary (in this case SARS-CoV-2) infection and the bystander infection. The bystander infection supposedly acts essentially as an adjuvant to provoke non-specific up-regulation of innate immunity creating the hyperinflammatory environment in which self-tolerance can be abrogated. Thus, the bystander theory leaves unresolved why only a small and very select set of pathogens were found here to be highly associated both as mimics of COVID-19 TCR sequences and as co-infections in COVID-19. Why, in short, should SARS-CoV-2 seem to require not just any bystander infection but very particular ones? Additionally, the bystander theory still leaves unaddressed the observation that TCR sequences mimic the bacterial agents associated with disease and, like the original anti-idiotype theory, cannot explain the mimicry of TCRs for host autoantigens. 

The final possible explanation for the results reported here not only integrates the basic concepts involved in the previous three theories but also predicts the TCR mimicry of complementary sets of pathogen antigens and host autoantigens that remains unexplained by them. This fourth explanation is that autoimmune disease is triggered by specific pairs of pathogens that present sets of complementary antigens [119,120,121,122,123,124,125,126,127,128,129,130,131]. In the complementary antigen theory, each antigen mimics host autoantigens that are, in turn, complementary to each other. This theory has previously been applied to understanding a number of autoimmune diseases that are of direct relevance to COVID-19 complications, including type 1 diabetes [9], autoimmune coagulopathies [69,70,121], autoimmune myocarditis [73,107,108,122], and anti-neutrophil cytoplasmic antibody (ANCA)-associated vascular autoimmune diseases [124,125,126,127,128,129,130,131]. The response of the immune system to such complementary antigens would be to produce sets of complementary TCRs that would have the same relationship to each other as the idiotype–anti-idiotype TCR pairs that would result from the anti-idiotype theory (Figure 13); however, in this instance, each of the TCR pairs would be produced as a primary idiotypic response to one of the complementary antigens. One of these antigens would derive from SARS-CoV-2; the other from one of the small set of bacterial co-infections identified by expanded TCRs that mimic autoantigens. Thus, this complementary antigen theory differs from the bystander activation theory, which permits any adjuvant-like cofactor infection to play the role of increasing inflammation, instead requiring that a co-infection or super-infection of SARS-CoV-2 must be antigenically complementary to the viral antigens. It follows that while many other viruses (e.g., adenoviruses, respiratory syncytial virus, influenza viruses, rhinoviruses, etc.), bacteria (*Clostridia*, *Legionella*, *Mycoplasmas*), and fungi or yeast (e.g., *Candida*, *Aspergillus*) have been found co-infecting COVID-19 patients, and might be expected to act as bystander infections to increase inflammation, only a select subset of microbes (e.g., *Streptococci*, *Staphylococci*, *Klebsiella*, and *Enterococci*) present antigens complementary to SARS-CoV-2 that can act as triggers of specific types of autoimmune disease. The complementary antigen theory also predicts that different combinations of these virus–bacteria pairs will result in different autoimmune complications depending on the sets of host mimicries expressed dominantly by the microbial pair. If the virus–bacterium pair mimic heart proteins, then autoimmune myocarditis may result; if platelet, fibrin or red blood cell antigens, then coagulopathies; if vascular antigens, MIS-C or KD. The otherwise unexplained mimicry of the TCRs from COVID-19 patients for SARS-CoV-2 follows from the fact that the bacterial antigens are complementary to SARS-CoV-2 so that TCRs induced against the bacterial antigens mimic the SARS-CoV-2 antigens. Conversely, the complementarity of the antigens means that TCRs expanded by stimulation by SARS-CoV-2 will identify their complementary bacterial antigens. 

Three predictions of the complementary antigen theory distinguish it from the other theories. One is the prediction that antigens exist on SARS-CoV-2 and its primary bacterial co-infections in COVID-19 that are complementary to each other. This complementarity has been demonstrated experimentally by showing that polyclonal antibodies against SARS-CoV-2 whole virus, or its spike protein, bind specifically and with high (nanomolar) affinity to polyclonal antibodies against several bacteria including group A *Streptococci*, *Staphylococcus aureus* and *Klebsiella pneumoniae* [69,70]. *Enterococci* were not, unfortunately, tested in these studies. Figure 14 summarizes studies demonstrating that such complementarity between viral and bacterial antibodies is rare.

A second unique test of the complementary antigen theory that differentiates it from the other theories is the prediction that induction of disease requires pairs of microbes that induce TCRs or antibodies that each mimic human autoantigens. The data supporting this prediction were summarized above in Figure 10 which illustrates the fact that the range of autoantibodies found in COVID-19 coagulopathies can only be explained by responses to SARS-CoV-2 and at least one bacterium. Patients who develop COVID-19 coagulopathies are characterized by the presence of multiple autoantibodies against blood-related autoantigens, including cardiolipin (CL), beta 2 glycoprotein I (β2GPI), platelet factor 4 (PF4) and usually one or more coagulation factors such as Factor 2 (prothrombin), von Willebrand Factor (vWF), Fact VIII and/or Factor X, whereas patients testing positive for only one of these autoantibodies do not develop coagulopathies (reviewed in [69,70]). Notably, antibodies against SARS-CoV-2 do not recognize either CL or β2GPI and cannot therefore account for the production of autoantibodies in these patients; however, antibodies against group A *Streptococci*, *Staphylococci*, *Klebsiella pneumoniae* and *E. coli* do recognize CL and β2GPI making them possible inducers of these autoantibodies (Figure 9) [69,70]. On the other hand, SARS-CoV-2 antibodies do recognize PF4, prothrombin and thrombin, and vWF, whereas antibodies against bacteria very rarely do so (Figure 9) [69,70]. Thus, to obtain the mix of autoantibodies that characterizes COVID-19 patients who develop coagulopathies, it is very likely that both SARS-CoV-2 and a bacterial co-infection with an appropriate bacterium such as *Streptococcus*, *Staphylococcus*, *Klebsiella* or *E. coli* is necessary. 

Similarly, patients who develop vascular and myocardial autoimmunity following COVID-19 are characterized by displaying antibodies that cross-react with cardiac cardiolipin (CL), alpha and beta adrenergic receptors, as well as myosin and collagen [132,133]. As with COVID-19 coagulopathies, SARS-CoV-2 antibodies do not recognize CL (Figure 10), requiring a bacterial source, such as *Streptococci,* to induce these antibodies, while no bacterium thus far tested induces antibodies that cross-react with adrenergic receptors while antibodies against the SARS-CoV-2 spike protein do. Thus, once again, the combination of autoantibodies present in COVID-19 patients with autoimmune myocarditis and vasculitis seems to result from a combined SARS-CoV-2–bacterial infection.

A third unique test of the complementary antigen theory that differentiates it from the other theories involves the prediction that the targets of autoimmunity should, like the inducing antigens, be themselves complementary *to each other*. This is certainly the case. Figure 15 summarizes the binding interactions known to occur between the various blood, extracellular matrix, and muscle proteins that are targets of autoimmune disease processed in COVID-19 complications (reviewed in [70]). In muscle- and vascular-related autoimmunity, laminins, collagens and keratins bind together to form the extracellular matrix while actin and myosin form the complex actinomyosin. CL binds to a range of phospholipid-binding proteins including phosphodiesterases, b2GPI, PF4, and vWF. vWF, in turn, binds to several other blood coagulation factors, and so on. Thus, the targets of autoantibodies found in COVID-19 and MIS-C patients with these autoimmune complications are certainly complementary autoantigens.

In sum, the only autoimmune disease theory that currently predicts that TCR sequences expanded during the disease process will mimic the antigenic sequences of the triggering agents as well as the host autoantigen targets of the disease is the complementary antigen theory. 

There is, however, one final interpretation of the results reported here that follows not from any autoimmune disease theory but rather from Jerne’s anti-idiotype theory of immunological control [134]. In Jerne’s theory, eliciting idiotypic antibodies or T cells leads several weeks later to the production of anti-idiotypes that regulate the idiotypic response after it has eliminated the initiating antigenic challenge. Two scenarios might follow. One is that a SARS-CoV-2 infection induces an anti-idiotypic immune response. As a consequence of the complementarity just established in discussing the complementary antigen theory, the resulting anti-idiotypic TCRs would be likely to mimic some of the bacterial infections to which SARS-CoV-2 predisposes. The existence of such anti-idiotypic TCRs mimicking these bacteria might then inhibit an immune response to them resulting in increased susceptibility to bacterial infection. Conversely, people who have been infected with one or more of these bacteria prior to exposure to SARS-CoV-2 might have anti-idiotypic bacterial TCRs that mimic SARS-CoV-2. The existence of these anti-idiotypic SARS-CoV-2 mimics might inhibit the immune response to the virus, resulting in an increased susceptibility to severe viral infection. In either case, the probability of increased susceptibility to, and/or severity of, disease might increase the probability of subsequent autoimmune complications. It is important to emphasize that the difference between this Jerne-network theory-based explanation for TCR amplification in COVID-19, MIS-C and KD patients differs from the complementary antigen theory mainly in terms of the timing of the viral and bacterial infections. If the viral and bacterial infections overlap in time, then the TCRs elicited will all be idiotypic (even though some may be complementary); whereas, if one of the infections precedes the other by sufficient time to elicit anti-idiotypic TCRs (i.e., separated by at least several weeks), then the Jerne-network explanation would be more likely. 

### 3.3. Further Tests of the Theories

Further studies and tests are required to differentiate the various theories from each other. Starting with the Jerne-network theory, one novel test would be to determine whether animals exposed to SARS-CoV-2 antigens to a degree sufficient to elicit anti-idiotypic TCR (or antibody) responses become more susceptible to the bacterial infections identified here as being possibly complementary (e.g., *E. faecium*, *Streptococci*, *Staphylococci*, etc.). Conversely, do animals exposed to these bacteria to a degree sufficient to elicit anti-idiotypic TCR (or antibody) responses become more susceptible to SARS-CoV-2 (or other viruses). Additionally, it would be interesting to know whether the anti-idiotypic TCRs correspond to sequences identified in this study as SARS-CoV-2-like or bacteria-like. Evidence of such a correspondence would help to demonstrate the complementarity of some of these TCRs for each other, while the absence of such a correspondence would argue against the complementarity of the antigens. However, such data would not distinguish between the Jerne-network theory and the complementary antigen theory without further tests to determine whether the combined infections (SARS-CoV-2 plus one of the identified bacteria) can elicit pairs of idiotypic TCR sets that act like idiotype–anti-idiotype pairs. 

Tests of the various autoimmune disease theories against each other are also possible. One would consist of inoculating susceptible experimental animals, such as golden hamsters, with SARS-CoV-2 by itself, with the individual bacterial and viral agents associated with severe COVID-19 cases, and with combinations of SARS-CoV-2 and these bacteria or viruses. Particularly promising combinations would involve SARS-CoV-2 with a group A *Streptococcus,* such as *S. pneumoniae* or *S. pyogenes,* as a possible model for autoimmune myocarditis; SARS-CoV-2 with *Staphylococcus aureus* or *haemolyticus* as a model for autoimmune coagulopathies; SARS-CoV-2 with *Enterococcus faecium* as a possible model for MIS-C; and one of the coronaviruses such as the HKU serotype with *E. faecium* as a model for KD. Single-agent models such as the molecular mimicry theory and the anti-idiotype theory would predict that autoimmunity might result with the individual microbes whereas the bystander infection model and complementary antigen models would predict that the combination of microbes will be necessary. The bystander theory can, in turn, be differentiated from antigenic complementarity by the range of microbes that can be substituted for each other to induce autoimmune disease. 

Additionally, the TCRs can themselves be used as experimental agents. It has, for example, been demonstrated using synthesized TCR sequences that TCRs induced in diabetes are complementary to each other as well as to their autoantigen targets [8,9]. Sets of the TCRs identified as SARS-CoV-2 mimics and sets of TCRs identified as bacterial mimics could be synthesized and tested for the recognition of each other and of the various autoantigens identified as possible mimics and targets. Such studies can be done with methods such as ultraviolet spectroscopy, mass spectrometry, nuclear magnetic resonance spectroscopy, etc. [8,9]. Alternatively, T cells specific for SARS-CoV-2 or for the bacteria identified here could be isolated and tested to determine whether they recognize each other as idiotype–anti-idiotypes. The existence of idiotype–anti-idiotype pairs of TCRs in COVID-19 autoimmune diseases can be considered a prediction of both the anti-idiotype and complementary antigen theories and a further test. 

### 3.4. TCR Sequences as Clues to the Causes of Autoimmune Diseases and Their Specific Treatment

Regardless of the explanation for the TCR mimicry of pathogen antigens and host autoantigens, the most important implication of these results is the possibility that the triggers of autoimmune diseases might be derived from such mimicry. With a large enough database of the distribution of randomly occurring TCR–microbe mimicry against which to compare, it might be possible to perform the type of analysis carried out here for groups of individuals sharing a common autoimmune disease and to determine the most probable microbial trigger(s) of that disease. It might even be possible, as Figure 5, Figure 7B, Figure 8 and Figure 9 and the Appendix A and Appendix B suggest, that the expanded TCRs from individual patients may be sufficient to identify the triggers of their specific autoimmune disease. Such knowledge might permit novel treatments tailored to blocking the TCRs mediating the disease to be developed using, for example, antisense techniques. 

Presumably, the analysis of antibody sequences derived from autoimmune diseases, such as KD [135,136,137], would yield similar or identical results in terms of microbial and host autoantigen matches to those derived from TCRs, providing another way to test the results reported here. Such analyses might also expand the possible treatments for autoimmune diseases beyond cell-mediated immunity to mediating disease-specific antibodies as well. 

Most importantly, these results suggest that the primary reason for the failure of some seventy years of research to reveal the cause of any human autoimmune disease may have been the search for single causal agents. If specific pairs or sets of microbes are necessary to trigger any particular autoimmune disease, then epidemiological and etiological studies must be conducted in novel ways that can identify patients experiencing combined infections. Such a combination-based explanation of autoimmunity would also go a long way towards helping to explain how a single agent, such as SARS-CoV-2, might be able to induce many different autoimmune diseases depending on the particular virus, bacterium or fungus with which it is paired. 

### 3.5. Implications for the Prevention of COVID-19-Associated Autoimmune Syndromes

One of the most important implications of the present study is found in the possibility that the autoimmune complications that characterize post-COVID-19 syndromes such as the so-called “long COVID” may largely be preventable. Beyond the obvious protection offered by COVID-19 vaccines, one preventative measure would be to optimize immunity against *Streptococcal* and *Haemophilus influenzae*, *type B* (Hib) infections by means of pneumococcal and Hib vaccinations. Studies involving hundreds of thousands of people have consistently reported that groups with high rates of pneumococcal and Hib vaccination are significantly less likely to develop severe COVID-19 or die from it than groups with low rates [138,139,140,141,142,143,144,145,146,147]. The synergistic activity of bacteria for which there are no current vaccines, such as *Staphylococci* and *Enterococci*, might be blunted by routine testing for infections, timely antibiotic use or even prophylactic use of antibiotics among high-risk patients. On this point, it is notable that the severity of COVID-19, which is associated with the risk of post-COVID-19 complications, such as autoimmunity, can be moderated by treatment with antibiotics prior to admission to intensive care or exacerbated if treatment for bacterial co-infection is delayed to the mid-to-late phase of the disease [148].

### 3.6. Limitations of the Study

Several limitations are inherent in the methods utilized in this study. One limitation of this study was that it utilized mainly aggregates of very small sets of TCR sequences that had been highly expanded in individual patients. On the one hand, the use of these data ensured that the TCR sequences were from the most highly activated T cells in the patients; on the other hand, there is no way to know what the optimal number of range of sequences to analyze and therefore to predict what may have been missed or included unnecessarily. 

Another important limitation of the study was that the TCR sequences were universally derived days or weeks following the onset of COVID-19, MIS-C or KD at a single time-point. As a consequence, it is impossible to rule out the possibility that the presence of expanded TCR subsets preceded COVID-19 and may have played a role in predisposing individuals to severe infections and subsequent autoimmune diseases. On this point, since this paper was first submitted, pronounced skewing of TCR sequences towards expansion of TRBV11-2 chains with high junctional and CDR3 diversity among MIS-C patients observed here has also been observed in a much broader study of MIS-C TCRs compared with TCRs recovered from both mild COVID-19 and healthy individuals [149]. Whether such skewing is a result or a predisposing cause of MIS-C susceptibility is a question of great importance because if it is predispositional, then children at greatest risk for MIS-C might be identified in advance of infection. If similar skewing pre-dates severe COVID-19 in adults, they, too, might benefit from being identifiable prior to developing complications. Longitudinal studies are clearly needed and while these would optimally be performed in human patients, animal models may be much more easily amenable to such studies. 

Perhaps the most important limitation of this study is that it was necessary to use published sets of TCR sequences that were often small and some of which aggregated data from many patients so that it was not possible to analyze TCR similarities for all of the individuals. There is no doubt that larger sets consisting of TCR sequences from many more individuals would help to validate or invalidate the results reported here. For example, a very large set of TCR sequences from healthy individuals can be found in [126] and COVID-19 TCRs in [127]. Harking back to the first limitation, however, there may be a danger in using sets of TCRs that include hundreds of TCRs from single individuals in that the important information required to identify microbial triggers and host autoantigens might be swamped out by large numbers of sequences irrelevant to the disease. Thus, while larger numbers of individuals sets of TCR sequences are probably very important to obtain, these sets should probably be limited to highly expanded TCRs associated with the particular disease (and its antigenic targets) being studied. Larger sets of data would undoubtedly resolve whether some of the not-quite-statistically significant observations are “real” or not.

A fourth limitation of this study is that the analysis of TCR sequence similarity was done by hand, which limited the number of sequences that could be handled in a reasonable amount of time and is probably prone to a certain amount of investigator error that might be avoided by automation. Clearly a future study of this type would benefit greatly from being computerized so that not only could much larger numbers of TCR sequences be explored but also proteomic databases of viruses, bacteria, fungi and human antigens more deeply mined. It is quite possible that some important microbial mimics and autoantigens were missed by the limited ranges used in performing the current study. Automation would also make it much easier to subject the results to subset analyses to determine whether, in KD for example, there are some sets of individuals whose disease results from a coronavirus–*Enterococcus* combination and others whose diseases are associated with some other (at this point, hypothetical) virus–bacterium or virus–virus combination. Such information would have been swamped out by the aggregate method utilized here. 

Finally, it is possible, though highly improbable, that the results reported here are entirely artifactual due to contamination of the TCR sequences by viral or bacterial sequences. Such contamination would be extremely unlikely since all of the studies from which the TCR sequences were derived (see sources listed in Section 4 below) utilized DNA primers designed to recognize highly conserved, genetically encoded TCR sequences immediately preceding the V-D-J regions that were sequenced. The viruses and bacteria that are over-represented in our analysis would have to have mimicked not only the variable regions reported here but have been *identical* to a much longer region preceding this variable region as well. While theoretically possible, there is no evidence for such highly conserved identities. Nature might, of course, surprise!

In short, this is a pioneering effort with all of the limitations that the first explorations inevitably have, and subsequent studies will undoubtedly find ways to do the type of analysis trialed here using better methods. 

## 4. Materials and Methods

### 4.1. Similarity Searches

Similarity searches comparing TCR sequences with virus, bacteria and human proteins were carried out using the standard protein BLASTp (protein–protein similarities) at the National Center for Biotechnology Information (NCBI) at the National Library of Congress, Washington, DC, USA. (https://blast.ncbi.nlm.nih.gov/Blast.cgi?PAGE=Proteins, accessed between 1 January 2020 and 1 November 2022). Each TCR sequence was input as a FASTA sequence; the UniProtKB/SwissProt database was selected with an appropriate organism limitation (viruses, taxid 10239; bacteria, taxid 2; homosapiens, taxid 9606). The general parameters were set with 250 sequences to display; threshold at 0.5; initiating word size, 2; BLOSSUM80; and filtering for low-complexity regions. The human matches were limited to E < 101 after the search was completed so as to ensure high-quality matches. The resulting matches were hand-curated to identify the approximately 40 viruses, bacteria and human proteins analyzed in the figures presented in this study. Selection of these particular viruses and bacteria was based on a previous study [8] in which their similarity profiles were evaluated in terms of type 1 diabetes and Crohn’s disease. The human proteins chosen for analysis were chosen in terms of their likelihood of being involved in coagulopathies [69,70], myocardial [119,120] or vascular [128,129,130] or olfactory/taste [150] autoimmune diseases associated with COVID-19 and included a number of “negative control” proteins such as keratins, mucins and tropomyosin that were not expected to be targets. 

### 4.2. TCR Sources

Normal TCR Sources: 100 randomly selected entries from [151] and: [8,152]. 

COVID-19 TCR Sources: [65,153,154,155,156]. 

MIS-C TCR Sources: [47,157]. 

KD TCR Sources: [80,158]. 

### 4.3. Statistics

A chi-squared test (http://www.quantpsy.org/chisq/chisq.htm) was used to make pair-wise comparisons between the percentage of matches for TCRs to the set of human viruses, bacteria, and proteins selected for analysis (see above) for the COVID-19, the MIS-C, and the KD groups. Because multiple chi-squared tests were run for each TCR group, a Bonferroni correction was applied to the resulting *p* values (http://www.winsteps.com/winman/bonferroni.htm). Because no significant difference was demonstrated between the percentage or overall distribution of the healthy TCR group as compared with randomized TCR sequences and antisense TCR sequences [8], it was assumed that this group could be used here as well as a reasonably randomized set of TCRs for statistical purposes. 

## 5. Conclusions

This paper reports the unexpected observation that about a quarter of highly expanded TCR sequences derived from severe COVID-19 and MIS-C patients mimic SARS-CoV-2 protein sequences and, similarly, the same percentage of TCR sequences derived from KD patients mimic proteins from other coronaviruses. An additional surprise was that statistically significant proportions of these TCR sequences also mimicked the proteins specifically from bacteria highly associated with COVID-19 and KD as co- or super-infections. These surprising results suggest that TCR sets expanded in pairs or combinations of viral and bacterial infections that trigger the autoimmune diseases. Additionally, the expanded TCR sets mimic to a statistically significant degree the main autoantigenic human proteins targeted by each autoimmune complication. These results are predicted by only one theory of autoimmune causation, which is the complementary antigen theory. If this theory is correct, then sequencing of TCRs in autoimmune diseases should be able to identify the specific triggers of each disease and may provide sufficient information to devise specific treatments to impair the activity of these particular TCR-bearing T cells. The information may additionally be validated or invalidated by examining the hypervariable regions of antibody sets stimulated in autoimmune diseases and such information may provide the basis for setting up new types of animal models for autoimmune diseases based on the actual triggers involved in human pathogenesis. 

## Figures and Tables

**Figure 1 ijms-24-01335-f001:**
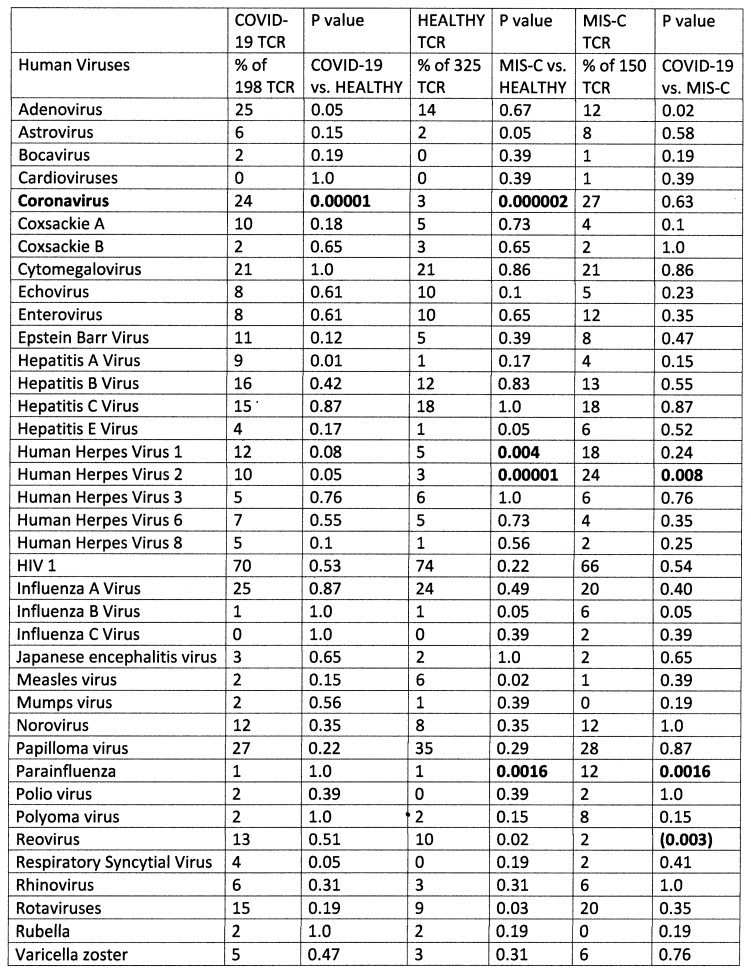
Summary of frequency (by percentage of TCRs tested) of TCR sequence similarities for hospitalized COVID-19 patients, healthy individuals, and multisystem inflammatory syndrome in children (MIS-C) patients to proteins from human viruses. The statistical significance of the differences between each pairing was determined by chi-squared analysis providing a *p* value. However, because each TCR was analyzed against every virus, a Bonferroni correction was required to interpret the resulting *p* values such that to reach a significance of *p* < 0.05 after the correction, the chi-squared value must be 0.002 or less. Values less than 0.002 or approaching it are bolded in the figure for ease of recognition.

**Figure 2 ijms-24-01335-f002:**
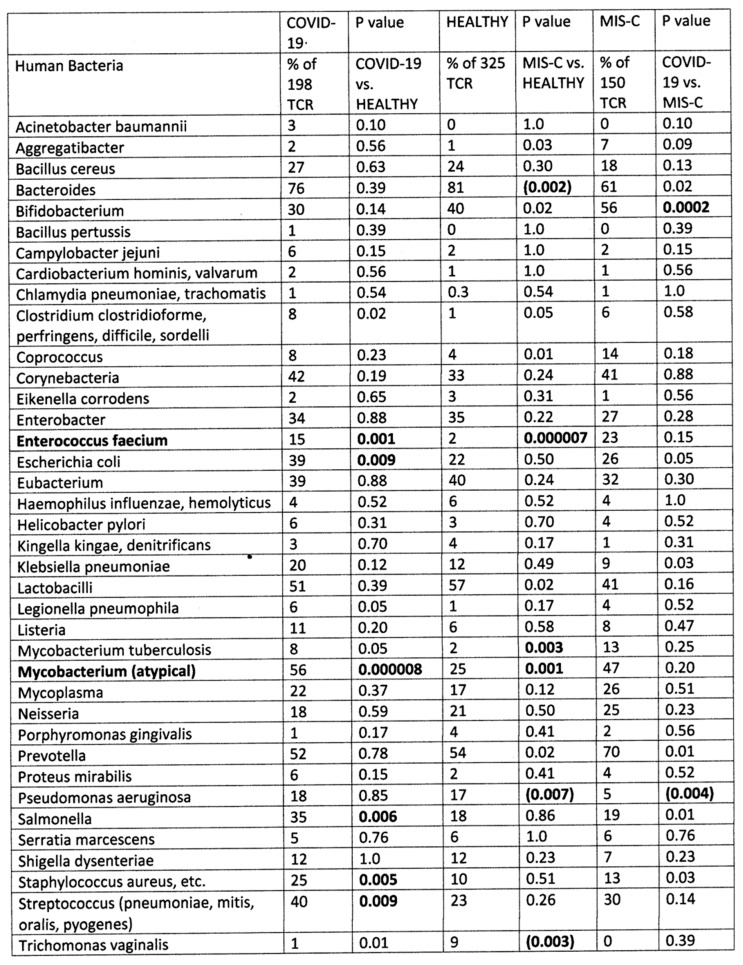
Summary of frequency (by percentage of TCRs tested) of TCR sequence similarities for hospitalized COVID-19 patients, healthy individuals, and multisystem inflammatory syndrome in children (MIS-C) patients to proteins from human bacteria. The statistical significance of differences between each pairing was determined by chi-squared analysis providing a *p* value. However, because each TCR was analyzed against every virus, a Bonferroni correction was required to interpret the resulting *p* values such that to reach a significance of *p* < 0.05 after the correction, the chi-squared value must be 0.002 or less. Values less than 0.002 or approaching it are bolded in the figure for ease of recognition. *p* values in parentheses indicate that there is a significant decrease in prevalence of matches compared with the healthy population.

**Figure 3 ijms-24-01335-f003:**
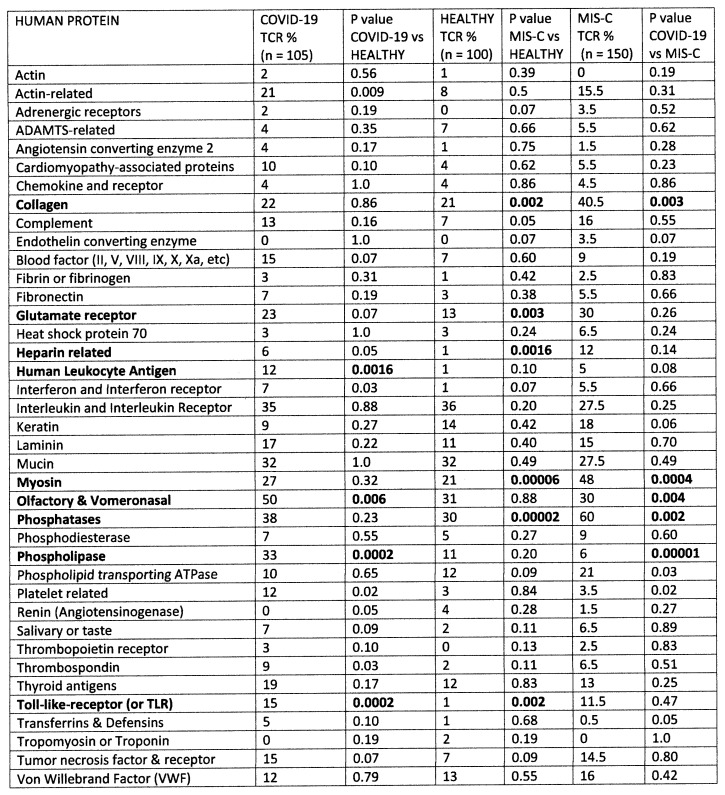
Summary of frequency (by percentage of TCRs tested) of TCR sequence similarities for hospitalized COVID-19 patients, healthy individuals, and multisystem inflammatory syndrome in children (MIS-C) patients to human proteins that may be targets of autoimmunity in these diseases. The statistical significance of differences between each pairing was determined by chi-squared analysis providing a *p* value. However, because each TCR was analyzed against every virus, a Bonferroni correction was required to interpret the resulting *p* values such that to reach a significance of *p* < 0.05 after the correction, the chi-squared value must be 0.002 or less. Values less than 0.002 or approaching it are bolded in the figure for ease of recognition.

**Figure 4 ijms-24-01335-f004:**
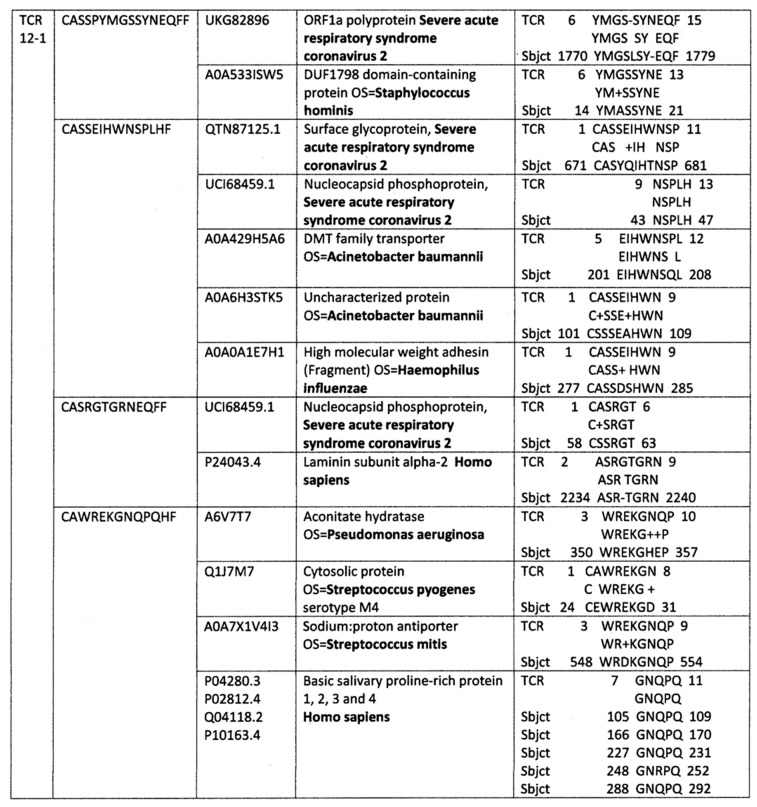
TCR sequences from patient 12-1 [74] and their similarities to viral, bacterial and human proteins found by BLAST. Not only does each TCR sequence mimic a virus, bacterium and/or human protein, the figure also illustrates that many of these viral, bacterial and human proteins mimic each other. Additionally, the specific human proteins identified by this analysis correspond with well-known targets of autoimmune processes associated with COVID-19 including cardiomyopathies (laminin) and anosmia/dysgeusia (basic salivary proline-rich protein). Additional individualized analyses can be found in Appendix A. Numbers in the second column from the left are the UNIPROT identifiers. Species names are in bold for ease of quick identification.

**Figure 5 ijms-24-01335-f005:**
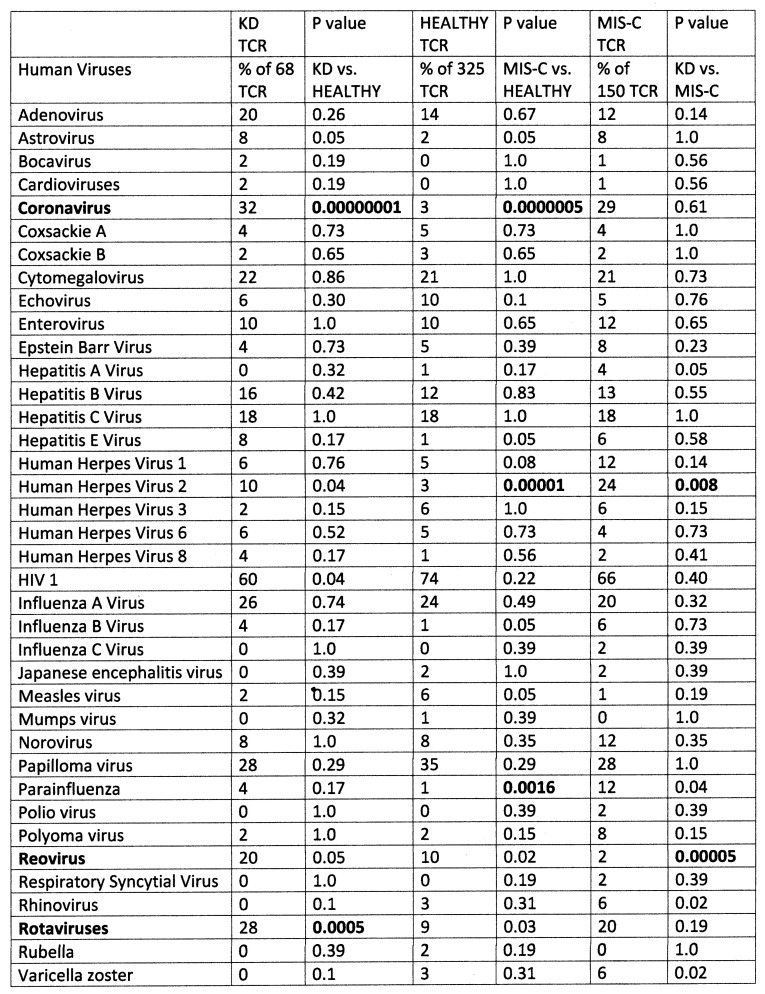
Summary of frequency (by percentage of TCRs tested) of TCR sequence similarities for Kawasaki disease (KD) patients, healthy individuals, and multisystem inflammatory syndrome in children (MIS-C) patients to virus proteins that may be targets of autoimmunity in these diseases. The statistical significance of differences between each pairing was determined by chi-squared analysis providing a *p* value. However, because each TCR was analyzed against every virus, a Bonferroni correction was required to interpret the resulting *p* values such that to reach a significance of *p* < 0.05 after the correction, the chi-squared value must be 0.002 or less. Values less than 0.002 or approaching it are bolded in the figure for ease of recognition.

**Figure 6 ijms-24-01335-f006:**
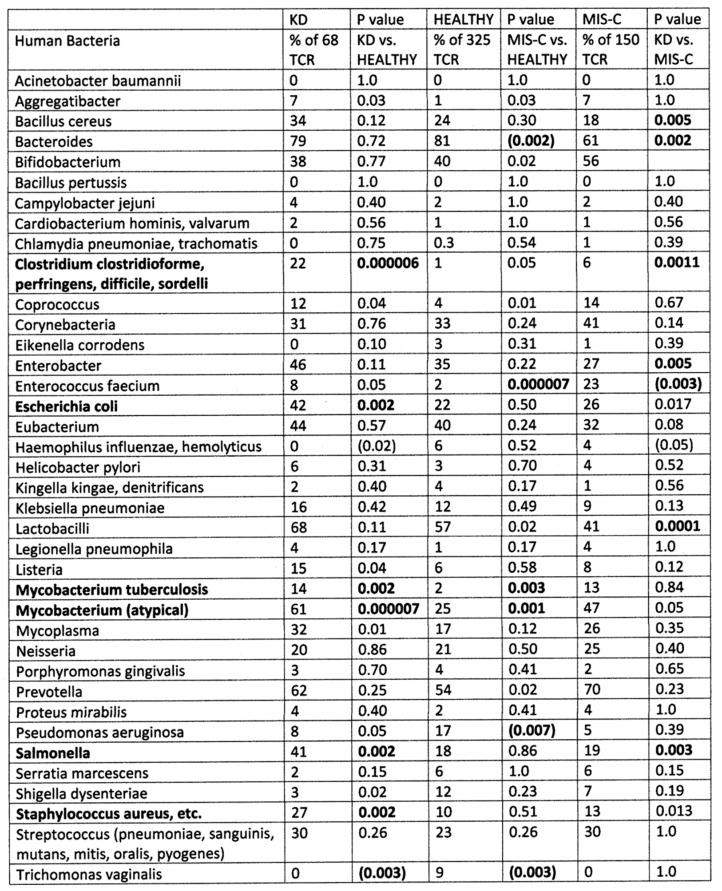
Summary of frequency (by percentage of TCRs tested) of TCR sequence similarities for Kawasaki disease (KD) patients, healthy individuals, and multisystem inflammatory syndrome in children (MIS-C) patients to bacterial proteins that may be targets of autoimmunity in these diseases. The statistical significance of differences between each pairing was determined by chi-squared analysis providing a *p* value. However, because each TCR was analyzed against every virus, a Bonferroni correction was required to interpret the resulting *p* values such that to reach a significance of *p* < 0.05 after the correction, the chi-squared value must be 0.002 or less. Values less than 0.002 or approaching it are bolded in the figure for ease of recognition. *p* values in parentheses indicate that there is a significant decrease in prevalence of matches compared with the healthy population.

**Figure 7 ijms-24-01335-f007:**
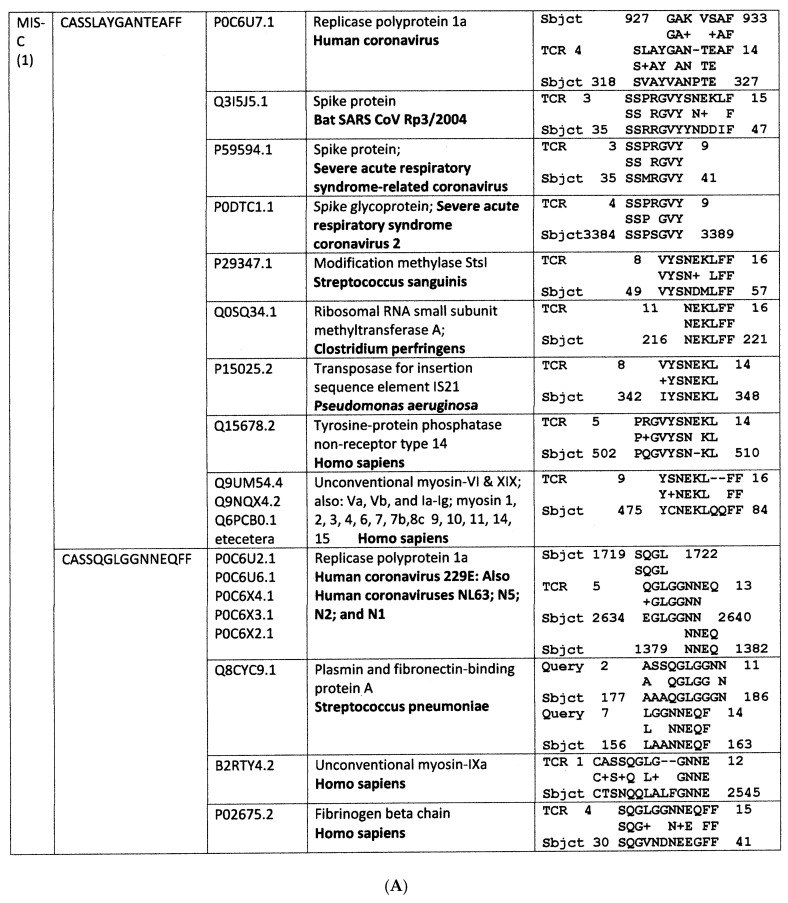
(**A**,**B**) Selected TCR sequences from a MIS-C patient [79] and their similarities to viral, bacterial and human proteins found by BLAST. Not only does each TCR sequence mimic a virus, bacterium and/or human protein, the figure also illustrates that many of these viral, bacterial and human proteins mimic each other. Additionally, the specific human proteins identified by this analysis correspond with well-known targets of autoimmune processes associated with MIS-C, including cardiomyopathies (myosins) and coagulopathies (von Willebrand Factor, fibrinogen and plasmin). Numbers in the second column from the left are the UNIPROT identifiers.

**Figure 8 ijms-24-01335-f008:**
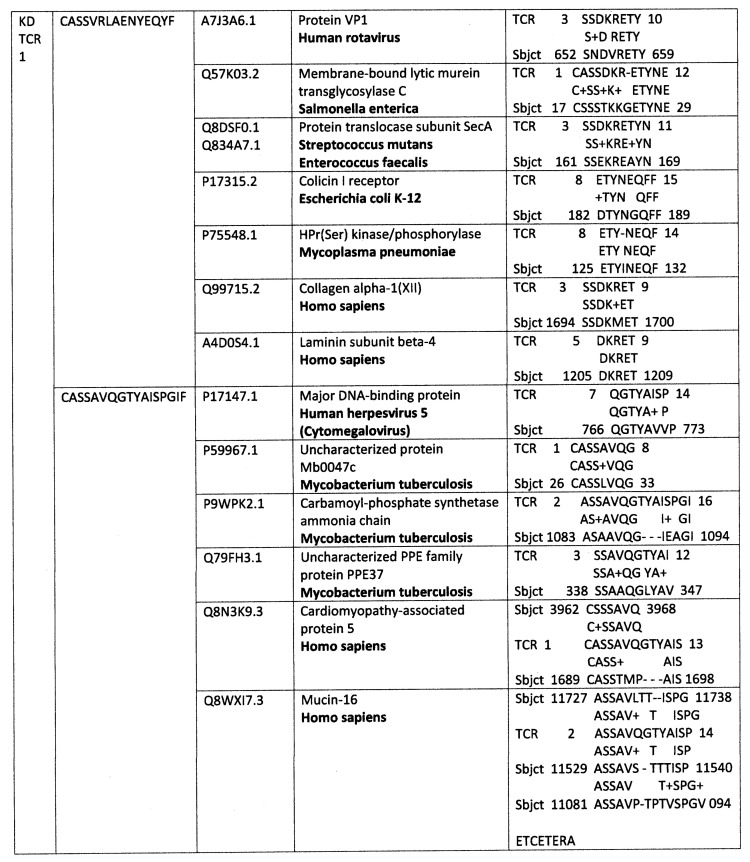
Selected TCR sequences from a Kawasaki disease (KD) patient number 1 [80] and their similarities to viral, bacterial and human proteins found by BLAST. Not only does each TCR sequence mimic a virus, bacterium and/or human protein, the figure also illustrates that many of these viral, bacterial and human proteins mimic each other. Additionally, the specific human proteins identified by this analysis correspond with well-known targets of autoimmune processes associated with MIS-C, including cardiomyopathies (myosins) and coagulopathies (von Willebrand Factor, fibrinogen and plasmin). Numbers in the second column from the left are the UNIPROT identifiers. Additional individual KD TCR mimicry examples are available in Appendix B.

**Figure 9 ijms-24-01335-f009:**
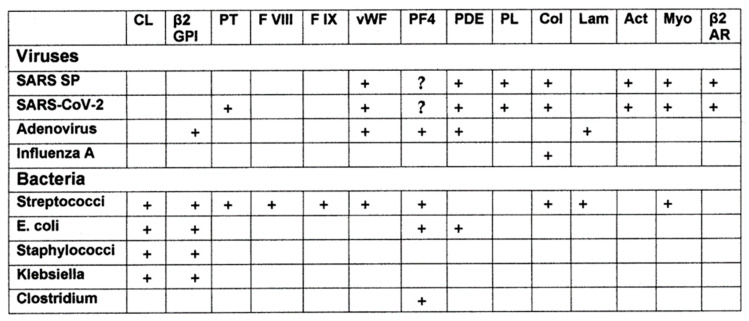
Summary of experimental results of binding to proteins targeted by autoimmune processes in COVID-19 by rabbit polyclonal SARS-CoV-2 antibodies, human anti-SARS-CoV-2 antibodies and similar antibodies against other infectious agents associated with COVID-19 summarized from [69,70,103,104,105,106]. Plus signs (+) indicated significant binding found between the antibody (left-hand column) and the human protein antigen (top row). Question marks (?) indicate that contradictory findings were reported by different studies, some observing significant binding while others reported no binding. CL = cardiolipin; β2GPI = beta 2 glycoprotein 1; PT = prothrombin; F VIII = factor VIII; F IX = factor IX; vWF = von Willebrand factor; PF4 = platelet factor 4; PDE = phosphodiesterase; PL = phospholipid; Col = collagen; Lam = laminin; Act = actin; Myo = myosin; β2AR = beta 2 adrenergic receptor. SARS SP = SARS-CoV-2 spike protein.

**Figure 10 ijms-24-01335-f010:**
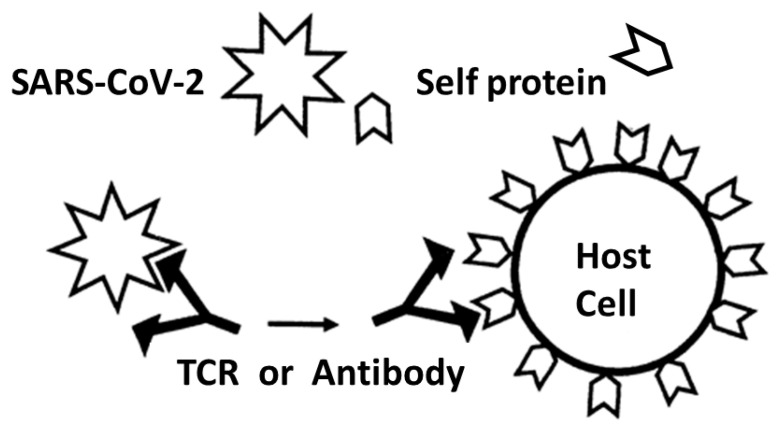
Schematic diagram of the molecular mimicry theory of autoimmune disease induction. A virus, such as SARS-CoV-2, mimics a self-protein on a host cell. The immune system responds by activating T or B cells that express T cell receptors (TCR) and/or antibodies (shown here for simplicity) that are complementary to the viral antigens. Because of the mimicry between the viral antigens and the self-protein, some of the resulting TCRs and/or antibodies may target host cells expressing these self-proteins, resulting in autoimmune disease [109,110,111,112]. While this theory is based on the sort of mimicry observed in expanded TCRs from COVID-19 patients, it actually predicts that the resulting TCR sequences should be complementary to SARS-CoV-2, not similar. Additionally, this theory makes no predictions that would explain TCR mimicry of the select set of bacteria that are found as co-infections in COVID-19.

**Figure 11 ijms-24-01335-f011:**
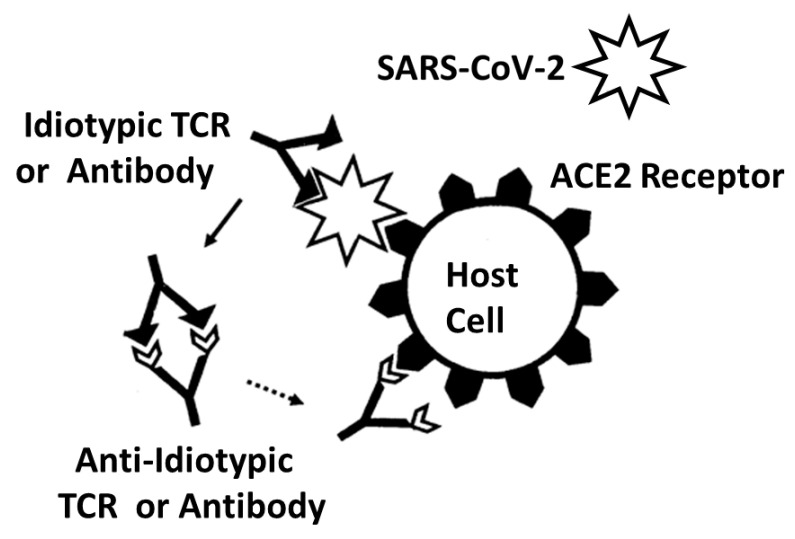
Schematic diagram summarizing the anti-idiotype theory of autoimmune disease induction [113,114,115]. In essence, a virus such as SARS-CoV-2 will induce a TCR and/or antibody idiotypic immune response (antibodies are illustrated for simplicity). If the idiotypic immune response is sufficiently robust, it may induce an anti-idiotypic response (solid arrow). The resulting TCRs or antibodies will then mimic the inducing antigen, in this case SARS-CoV-2, and target the same host cell receptors as does the virus (dotted arrow), in this case, the angiotensin-converting enzyme type 2 (ACE-2) receptor. This theory could explain how expanded TCR sequences mimic SARS-CoV-2 antigens but does not explain how these TCR sequences also mimic host autoantigens or their specific mimicry of bacteria known to co-infect COVID-19 patients.

**Figure 12 ijms-24-01335-f012:**
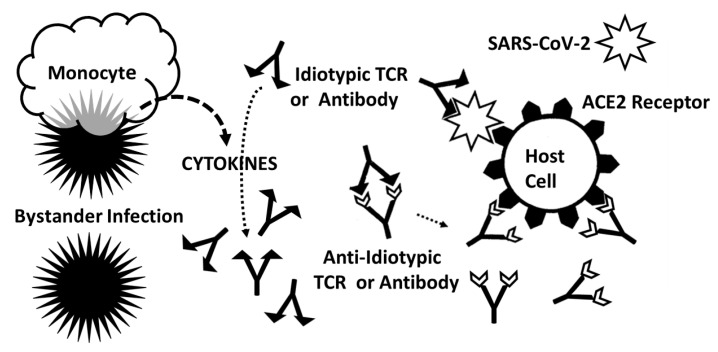
Schematic representation of the bystander activation theory of autoimmune disease [116,117,118]. The theory suggests that non-specific bystander infections stimulate a hyperinflammatory state in the innate immune system that results in over-production of cytokines (large dashed arrow from monocytes) enabling over-production of idiotypic TCRs and/or antibodies (dotted arrow from idiotypic TCR or Antiibody). The unusual production of idiotypic TCRs/antibodies then initiates the production of anti-idiotypes (small dotted arrow from anti-idiotypic TCR or antibody) that mimic the initiating microbes. This theory could explain both why only some individuals develop autoimmune diseases following COVID-19 (only those with co-infections do so) and also why the resulting TCRs mimic both SARS-CoV-2 and host antigens. It does not, however, explain why the bacterial mimicry observed here is limited to the most common co-infections found among COVID-19 patients.

**Figure 13 ijms-24-01335-f013:**
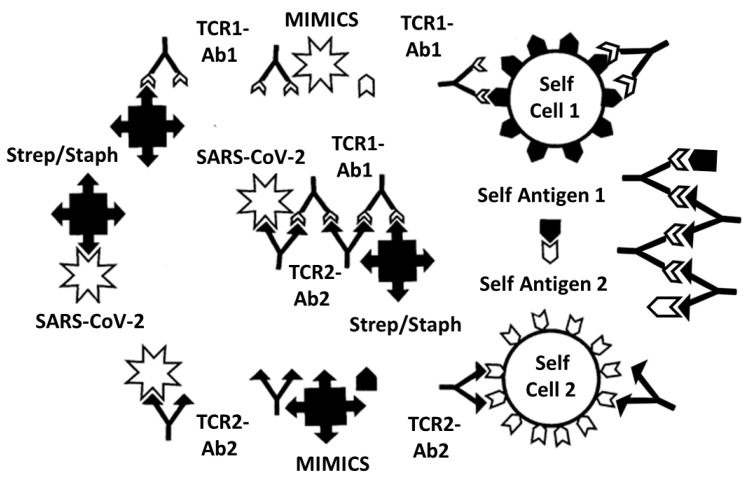
Schematic diagram summarizing the complementary antigen theory of autoimmune disease [119,120,121,122,123,124,125,126,127,128,129,130,131]. The theory proposes that autoimmune diseases are induced by pairs of infectious agents that express complementary antigens, in the case of COVID-19, SARS-CoV-2 and one of several specific bacteria such as *Streptococci* (Strep), *Staphylococci* (Staph), or *Enterococci*. Each microbe induces an idiotypic immune response (TCR or antibody—antibodies are shown here for simplicity) that is complementary to its antigen. Because the inducing antigens are themselves complementary, the resulting TCRs and/or antibodies will also be complementary, having an idiotype–anti-idiotype relationship (as in the anti-idiotype theory), but each produced in this case as an idiotypic response. As a consequence, each TCR/antibody will mimic one of the inducing antigens and, because each antigen mimics a host autoantigen, will also mimic a host antigen. The result will be the induction of TCRs/antibodies that bind to each other as well as to their respective microbes and to the host autoantigens that those microbes mimic. These relationships are exactly what is observed in the results reported here.

**Figure 14 ijms-24-01335-f014:**
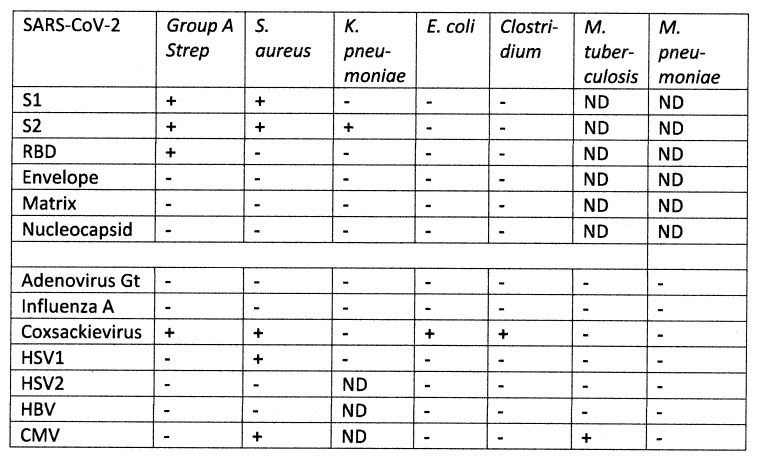
Summary of experimental studies of the binding of viral antibodies (left-hand column) to bacterial antibodies (top row) [69,70]; additional data from [20,21]. The plus signs indicate nanomolar binding between the antibody pair; minus signs indicate insignificant (micromolar or no observable binding) between the antibody pair. ND means that combination was not tested. S1, S2, RBD, envelope, matrix and nucleocapsid refer to specific proteins of the SARS-CoV-2 virus. Influenza A = influenza A virus; HSV = human herpes virus; HBV = hepatitis B virus; CMV = human cytomegalovirus; Strep = *Streptococci*.

**Figure 15 ijms-24-01335-f015:**
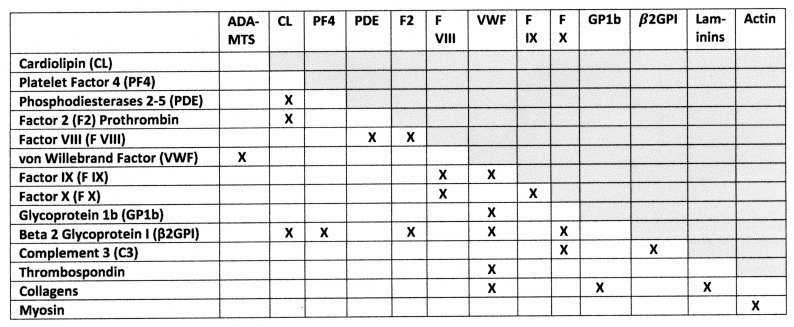
Summary of known binding interactions (i.e., autoantigen complementarity) reported among the various human protein targets of autoimmunity discussed in this paper (adapted from [70]). X indicates that the pair of proteins are known to bind to each other and thus display complementary regions. The abbreviations for the top row are provided in the left-hand column. Background shading blocks off duplicate entries.

## Data Availability

Original data sets and additional analyzed data are available by request from the corresponding author.

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
