# Peer review of "T Cell Receptor Sequences Amplified during Severe COVID-19 and Multisystem Inflammatory Syndrome in Children Mimic SARS-CoV-2, Its Bacterial Co-Infections and Host Autoantigens"

_ijms, 2023, doi:10.3390/ijms24021335_

Round 1

Reviewer 1 Report

Here, Root-Bernstein et al. utilize published TCR (T-cell receptor) data to suggest that the frequency of TCR and SARS-Cov-2 sequence similarities is very high in COVID-19 patients who also suffer from accompanying autoimmune and inflammatory complications. In the same data set, the authors also observe an elevated frequency of TCRs sharing their sequences with bacterial and host antigens. They posit that the mimicry between TCRs and SARS-Cov-2 antigens is a product of an idiotypic interaction in which bacterial and host antigens complementary to proteins of SARS-Cov-2 induced-generation of TCRs mirror SARS-Cov-2 proteins. In such circumstances, antibodies targeted against a SARS-Cov-2 virus would also bind to the TCR- mimics and an autoimmune disease would ensue.  The entire scenario they call, “complementary antigen theory of autoimmune disease”.

I think an interesting element (but not addressed) is the consideration of a therapeutic intervention: 1)  If the bacteria are part of the sequence leading to autoimmunity, then prophylactic antibiotics might, in select patients with known immune disorders, be useful to prevent long COVID. Any thoughts on the matter?

      2) Another unaddressed issue concerns how to investigate and eventually prove the hypothesis and its theoretical consequences.  In other words, comments should be made as to how experimentally substantiate the hypothesis through investigations.  In other words, a research proposal should be outlined that would advance the thesis.

Now for specific comments and questions:

1. Is the high abundance of SARS-Cov-2 protein sequences in the TCR sample really a product of T-cell mimicry or is it due to contamination by the virus. In fact, the authors deal with the data obtained from patients with severe COVID-19 infection. How are they certain that the data is completely free of viral sequences?

2. Can the authors rule out that T-cell receptor mimicry was present prior to the virus infection in these patients? If this was the case, the shared TCR and SARS-Cov-2 sequences could be responsible for a particular susceptibility to the subsequent COVID-19 infection and associated complications. More generally, there is a concern whether the conclusions are supported by the data. The data shows only a high frequency of TCR similarities to viral, bacterial, and host antigens in a group of COVID-19 patients. Absent are any causal links based on this limited (single time slice) evidence (see above).

3. The authors observe a high level of mimicry between TCRs and SARS-Cov-2 in MIC-S patients. They assume that these patients acquired this disease following SARS-Cov-2 infection, but at what time during the post-infection period did they develop the disease and what is the clinical data for these patients? Also, more information should be provided about the Kawasaki disease patients, in particular about when and in what clinical context the disease onset occurred after their SARS-Cov-2 infection. Answering these questions could possibly help to establish some casual links between the SARS-Cov-2 infection and the disease. 

4. A minor point concerns representation of the data, which is provided in the form of Excel spreadsheets with calculated exact frequencies, p-values, etc. These tables are named “figures,” but figures are missing in the “Results” section of the manuscript, which does not help to visualize the data. In addition, “figure 3” is missing (there is only figure 2 and 4) and the numbers of some figures seem mixed up in the manuscript. 

Author Response

Reviewer 1

Open Review

English language and style

( ) English very difficult to understand/incomprehensible
( ) Extensive editing of English language and style required
( ) Moderate English changes required
(x) English language and style are fine/minor spell check required
( ) I don't feel qualified to judge about the English language and style

Yes

Can be improved

Must be improved

Not applicable

Does the introduction provide sufficient background and include all relevant references?

(x)

( )

( )

( )

Are all the cited references relevant to the research?

(x)

( )

( )

( )

Is the research design appropriate?

(x)

( )

( )

( )

Are the methods adequately described?

(x)

( )

( )

( )

Are the results clearly presented?

(x)

( )

( )

( )

Are the conclusions supported by the results?

(x)

( )

( )

( )

Comments and Suggestions for Authors

Here, Root-Bernstein et al. utilize published TCR (T-cell receptor) data to suggest that the frequency of TCR and SARS-Cov-2 sequence similarities is very high in COVID-19 patients who also suffer from accompanying autoimmune and inflammatory complications. In the same data set, the authors also observe an elevated frequency of TCRs sharing their sequences with bacterial and host antigens. They posit that the mimicry between TCRs and SARS-Cov-2 antigens is a product of an idiotypic interaction in which bacterial and host antigens complementary to proteins of SARS-Cov-2 induced-generation of TCRs mirror SARS-Cov-2 proteins. In such circumstances, antibodies targeted against a SARS-Cov-2 virus would also bind to the TCR- mimics and an autoimmune disease would ensue.  The entire scenario they call, “complementary antigen theory of autoimmune disease”.

I think an interesting element (but not addressed) is the consideration of a therapeutic intervention: 1)  If the bacteria are part of the sequence leading to autoimmunity, then prophylactic antibiotics might, in select patients with known immune disorders, be useful to prevent long COVID. Any thoughts on the matter?

Yes, we have added a brief section on the use of bacterial vaccines such as Hib and Pneumococcal and prophylactic antibiotics.

      2) Another unaddressed issue concerns how to investigate and eventually prove the hypothesis and its theoretical consequences.  In other words, comments should be made as to how experimentally substantiate the hypothesis through investigations.  In other words, a research proposal should be outlined that would advance the thesis.

Actually, Section 3.3 already did this, but further experimental tests of the hypothesis are now added.

Now for specific comments and questions:

  1. Is the high abundance of SARS-Cov-2 protein sequences in the TCR sample really a product of T-cell mimicry or is it due to contamination by the virus. In fact, the authors deal with the data obtained from patients with severe COVID-19 infection. How are they certain that the data is completely free of viral sequences?

Interesting question. We cannot address it directly, since we did not derive the sequences ourselves, but we have added some additional detail on the methods used by our sources that make viral contamination unlikely. The unlikeliness stems from the use of specific DNA primers designed to recognize genetically encoded TCR sequences immediately preceding the V-D-J regions that are sequenced. The viruses and bacteria that are over-represented in our analysis would have to mimic not only the variable region but be identical to a much longer region preceding this variable region as well. While possible, this is extremely unlikely. We have, however, added this possibility to the limitations section at the end of the Disussion.

  1. Can the authors rule out that T-cell receptor mimicry was present prior to the virus infection in these patients? If this was the case, the shared TCR and SARS-Cov-2 sequences could be responsible for a particular susceptibility to the subsequent COVID-19 infection and associated complications. More generally, there is a concern whether the conclusions are supported by the data. The data shows only a high frequency of TCR similarities to viral, bacterial, and host antigens in a group of COVID-19 patients. Absent are any causal links based on this limited (single time slice) evidence (see above).

Another interesting possibility that we realized after submitting. We do not know of any way to rule out this possibility and have therefore added it in two places in the manuscript: one is as a possible additional explanation for the pathogenesis following our analysis of the various autoimmune disease theories and the other is as part of the limitations inherent in the study. This possibility clearly emphasizes the need for longitudinal studies of TCR arrays prior to infection and afterwards. Such studies could clearly be carried out in animals.

  1. The authors observe a high level of mimicry between TCRs and SARS-Cov-2 in MIC-S patients. They assume that these patients acquired this disease following SARS-Cov-2 infection, but at what time during the post-infection period did they develop the disease and what is the clinical data for these patients?

MIS-C is defined by presence of a positive test for SARS-CoV-2 (distinguishing it from Kawasaki Disease) and the studies we used clearly state that symptoms of MIS-C follow the virus infection by three or four weeks on average. We have clarified this point in the text and this fact is already stated in the various papers cited. In response to the other Reviewer, we have added additional references specifically on the point that MIS-C is an autoimmune disease, one criteria of which is the absence of active infection and demonstration of pathogenic antibodies and/or T cells.

Also, more information should be provided about the Kawasaki disease patients, in particular about when and in what clinical context the disease onset occurred after their SARS-Cov-2 infection. Answering these questions could possibly help to establish some casual links between the SARS-Cov-2 infection and the disease. 

Although KD symptoms overlap MIS-C, MIS-C is defined by preceding SARS-CoV-2 infection while the etiology of KD is not known at present. KD does not follow SARS-CoV-2 although it is thought to follow other viral or bacterial infections. Perhaps we did not make this point clear enough in the text, so we have now emphasized it.

  1. A minor point concerns representation of the data, which is provided in the form of Excel spreadsheets with calculated exact frequencies, p-values, etc. These tables are named “figures,” but figures are missing in the “Results” section of the manuscript, which does not help to visualize the data. In addition, “figure 3” is missing (there is only figure 2 and 4) and the numbers of some figures seem mixed up in the manuscript. 

We have labelled the tables as “Figures” in order to preserve the formatting that we have adopted. We have found from previous experience that journals always reformat tables, losing the key aspects of the presentation. We understand the confusion this causes for Reviewers but find that, in the long run, this strategy is effective for presenting the data in the form we want to have it presented.

The Figures referred to are not Excel spreadsheets. They are Word tables. Graphing the 120 entries that are contained in these tables results in unintelligible figures.

We apologize for the confusion in the Figure numbering and have corrected it. 

Submission Date

30 November 2022

Date of this review

11 Dec 2022 09:51:13

Reviewer 2 Report

-          Line 55, 169, 219, 451, 527, 562, 637, 726, 728, 748  there is extra space

-          Line 47, TCR was already defined previously.

-          Line 85, define MRI

-          Line 88, MIS-C was not considered as autoimmune unless stated otherwise

-          Line 128, there is an extra s letter in the sentence

-          Paragraph starting in the line 168. It should be clarified on the patients’ no besides TCR.

-          Page 7/50, line 261 should be Figure 3 not figure 4.  

-          Line 458 – mentioned that coronaviruses are common in KD – there are not so many studies to make this as a conclusion.

-          Line 470 the message is not clear. The subject of research is great. However, the figures are too many for the context and the message is not clear. 

Author Response

Reviewer 2

Open Review

English language and style

( ) English very difficult to understand/incomprehensible
( ) Extensive editing of English language and style required
( ) Moderate English changes required
(x) English language and style are fine/minor spell check required
( ) I don't feel qualified to judge about the English language and style

Yes

Can be improved

Must be improved

Not applicable

Does the introduction provide sufficient background and include all relevant references?

( )

(x)

( )

( )

Are all the cited references relevant to the research?

(x)

( )

( )

( )

Is the research design appropriate?

( )

(x)

( )

( )

Are the methods adequately described?

( )

(x)

( )

( )

Are the results clearly presented?

(x)

( )

( )

( )

Are the conclusions supported by the results?

(x)

( )

( )

( )

Comments and Suggestions for Authors

-          Line 55, 169, 219, 451, 527, 562, 637, 726, 728, 748  there is extra space

FIXED

-          Line 47, TCR was already defined previously.

REMOVED

-          Line 85, define MRI

DONE

-          Line 88, MIS-C was not considered as autoimmune unless stated otherwise

WE DON’T UNDERSTAND THIS COMMENT. MIS-C IS MOST CERTAINLY AUTOIMMUNE AND WE HAVE ADDED A SERIES OF REFERENCES SUBSTANTIATING THIS POINT. IF THE REVIEWER MEANT SOMETHING ELSE, WE DO NOT COMPREHEND.

-          Line 128, there is an extra s letter in the sentence

FIXED

-          Paragraph starting in the line 168. It should be clarified on the patients’ no besides TCR

NUMBER OF PATIENTS ADDED

-          Page 7/50, line 261 should be Figure 3 not figure 4. 

CORRECTED AND REST OF FIGURE NUMBERS ALSO CORRECTED

-          Line 458 – mentioned that coronaviruses are common in KD – there are not so many studies to make this as a conclusion.

NOW LINES 461ff: WE DON’T UNDERSTAND THE REVIEWER’S COMMENT. WE DID NOT SAY THAT CORONAVIRUSES ARE COMMON IN KD  THE POINT OF THE PREVIOUS PARAGRAPH IS THAT THERE ARE DIVERSE VIRUSES ASSOCIATED WITH KD. THE FACT THAT THE ONLY TYPE OF VIRUS THAT IS COMMON AMONG COVID-19 PATIENTS, MIS-C PATIENTS AND KD PATIENTS IS, HOWEVER, WORTH POINTING OUT, WHICH IS WHAT THIS PARAGRAPH DOES. NOTE THAT WE ARE CAREFUL TO POINT OUT THAT OTHER VIRUSES ARE IMPLICATED AS WELL SO THAT, IN CONTEXT, WE HAVE ALREADY MADE THE REVIEWER’S POINT.

-          Line 470 the message is not clear. The subject of research is great. However, the figures are too many for the context and the message is not clear. 

AGAIN, THE COMMENT IS TOO VAGUE TO ADDRESS. IS A PARTICULAR SENTENCE CONFUSING AND IF SO, IN WHAT WAY? OR IS THIS A GENERAL COMMENT ABOUT CONTEXT AND MESSAGING IN GENERAL (“THE FIGURES ARE TOO MANY…”)?  IN TERMS OF THE NUMBER OF FIGURES, THIS CRITICISM MAKES LITTLE SENSE TO US SINCE ANY CONCLUSIONS WE REACH ARE ONLY AS VALID AS THE DATA UPON WHICH THEY ARE BASED AND THE READER DESERVES TO HAVE AS MUCH OF THAT DATA AVAILABLE AS POSSIBLE. WE LEAVE THIS POINT UP TO THE EDITOR…

Submission Date

30 November 2022

Date of this review

28 Dec 2022 15:54:45